# tRNA genes rapidly change in evolution to meet novel translational demands

**Avihu H Yona[1†], Zohar Bloom-Ackermann[1†], Idan Frumkin[1†], Victor Hanson-Smith[2,3], Yoav Charpak-Amikam[1], Qinghua Feng[4], Jef D Boeke[5‡], Orna Dahan[1], Yitzhak Pilpel[1]\***

[1]Department of Molecular Genetics, Weizmann Institute of Science, Rehovot, Israel; [2]Department of Microbiology, University of California, San Francisco, San Francisco, United States; [3]Department of Immunology, University of California, San Francisco, San Francisco, United States; [4]Department of Pathology, University of Washington, Seattle, United States; [5]Department of Molecular Biology and Genetics, The Johns Hopkins University School of Medicine, Baltimore, United States

**Abstract** Changes in expression patterns may occur when organisms are presented with new environmental challenges, for example following migration or genetic changes. To elucidate the mechanisms by which the translational machinery adapts to such changes, we perturbed the tRNA pool of *Saccharomyces cerevisiae* by tRNA gene deletion. We then evolved the deletion strain and observed that the genetic adaptation was recurrently based on a strategic mutation that changed the anticodon of other tRNA genes to match that of the deleted one. Strikingly, a systematic search in hundreds of genomes revealed that anticodon mutations occur throughout the tree of life. We further show that the evolution of the tRNA pool also depends on the need to properly couple translation to protein folding. Together, our observations shed light on the evolution of the tRNA pool, demonstrating that mutation in the anticodons of tRNA genes is a common adaptive mechanism when meeting new translational demands.

**\*For correspondence:** Pilpel@weizmann.ac.il

†These authors contributed equally to this work

‡**Present address:** New York University Langone Medical Center, New York, United States

**Competing interests:** The authors declare that no competing interests exist.

**Reviewing editor**: Michael Laub, Massachusetts Institute of Technology, United States

## Introduction

The process of gene translation is fundamental to the function of living cells, and as such its apparatus is highly conserved across the tree of life (*Müller and Wittmann-Liebold, 1997*; *Itoh et al., 1999*; *Wolf et al., 2001*). Yet, the capacity of the translation machinery to adaptively evolve is crucial in order to support life in changing environments. Therefore, a key open question is to identify the mechanisms by which the translation machinery adapts to changing conditions.

A thoroughly studied aspect of translation that demonstrates its adaptation capacities is the different proportions by which synonymous codons are used, a phenomenon known as 'codon usage bias'. Although differential use of codons can be the result of neutral processes such as mutational biases and the genomic GC content (*Urrutia and Hurst, 2001*; *Rao et al., 2011*), natural selection also influences codon usage bias. Indeed, it has been demonstrated that codon choice affects expression level, protein folding, translational accuracy, and other translational features (*Akashi, 1994*; *Parmley and Hurst, 2007*; *Zhou et al., 2009*; *Hudson et al., 2011*). Since both neutral and selective processes govern codon usage bias, the balance between selection, mutational bias and drift is crucial in shaping the codon usage of each species (*Bulmer, 1991*). Importantly, although the selective advantage offered by alternative synonymous codons is considered to be moderate, it was recently demonstrated that selection can still shape codon usage patterns in vertebrates even with their small effective population sizes (*Doherty and McInerney, 2013*).

Notably, the differential usage of codons represents the evolution of the 'demand' aspect of translation, namely the codon usage of all expressed genes. Yet, the adaptation mechanisms of the 'supply',

**eLife digest** Genes contain the blueprints for the proteins that are essential for countless biological functions and processes, and the path that leads from a particular gene to the corresponding protein is long and complex. The genetic information stored in the DNA must first be transcribed to produce a messenger RNA molecule, which then has to be translated to produce a string of amino acids that fold to form a protein. The translation step is performed by a molecular machine called the ribosome, with transfer RNA molecules bringing the amino acids that are needed to make the protein.

The information in messenger RNA is stored as a series of letters, with groups of three letters called codons representing the different amino acids. Since there are four letters—A, C, G and U—it is possible to form 64 different codons. And since there are only 20 amino acids, two or more different codons can specify the same amino acid (for example, AGU and AGC both specify serine), and two or more different transfer RNA molecules can take this amino acid to the ribosome. Moreover, some codons are found more often than others in the messenger RNA molecules, so the genes that encode the related transfer RNA molecules are more common than the genes for other transfer RNA molecules.

Environmental pressures mean that organisms must adapt to survive, with some genes and proteins increasing in importance, and others becoming less important. Clearly the relative numbers of the different transfer RNA molecules will also need to change to reflect these evolutionary changes, but the details of how this happens were not understood.

Now Yona et al. have explored this issue by studying yeast cells that lack a gene for one of the less common transfer RNA molecules (corresponding to the codon AGG, which specifies the amino acid arginine). At first this mutation resulted in slower growth of the yeast cells, but after being allowed to evolve over 200 generations, the rate of growth matched that of a normal strain with all transfer RNA genes. Yona et al. found that the gene for a more common transfer RNA molecule, corresponding to the codon AGA, which also specifies arginine, had mutated to AGG. As a result, the mutated yeast was eventually able to produce proteins as quickly as wild type yeast. Moreover, further experiments showed that the levels of some transfer RNAs are kept deliberately low in order to slow down the production of proteins so as to ensure that the proteins assume their correct structure.

But does the way these cells evolved in the lab resemble what happened in nature? To address this question Yona et al. examined a database of transfer RNA sequences from more than 500 species, and found evidence for the same codon-based switching mechanism in many species across the tree of life.

namely the expression level of each tRNA type that is loaded with an amino acid, are not fully understood. While ribosomal genes do not exhibit appreciable changes in response to environmental alterations (*Müller and Wittmann-Liebold, 1997*; *Itoh et al., 1999*; *Wolf et al., 2001*), tRNA genes may provide an important source of evolutionary plasticity for fine tuning translation.

tRNAs constitute a fundamental component in the process of translation, linking codons to their corresponding amino acids (*Widmann et al., 2010*). tRNA genes are classified into gene families according to their anticodon, with each gene family containing between one and several copies scattered throughout the genome. Importantly, it has been experimentally observed for *Saccharomyces cerevisiae* (*Tuller et al., 2010*) and *Escherichia coli* (*Dong et al., 1996*) that the cellular concentrations of each tRNA family in the cell (i.e., the tRNA pool) correlate with its genomic tRNA copy number (*Percudani et al., 1997*; *Kanaya et al., 1999*). Notably, the rate-limiting step of polypeptide elongation is the recruitment of a tRNA that matches the translated codon (*Varenne et al., 1984*). Thus, the translation efficiency is defined as the extent to which the tRNA pool can accommodate the transcriptome (*Sharp and Li, 1987*; *Dos Reis et al., 2004*; *Stoletzki and Eyre-Walker, 2007*), thereby affecting protein production and accuracy.

In general, highly expressed genes exhibit a marked codon usage bias toward 'optimal' codons, whose corresponding tRNA gene copy number is high (*Sharp and Li, 1986a*, *1986b*). The evolutionary force that acts to maintain optimal translation efficiency of such genes was coined 'translational

selection' (*Dos Reis et al., 2004*). It was previously suggested that translational selection acts to maintain a balance between codon usage and tRNA availability. On the one hand, there is a selective pressure to increase the frequency of preferred codons in highly expressed genes. On the other hand, changes in the tRNA pool may also occur, for example duplication of tRNA genes for which high codon demand exists. Thus, codon frequencies and tRNA copy numbers coevolve toward a supply versus demand balance that facilitates optimal protein production (*Higgs and Ran, 2008*; *Gingold et al., 2012*).

The fitness effects of an unmet translational demand and its potential role in shaping the tRNA pool are not fully characterized. Evolutionary changes to the tRNA pool were appreciated mainly via bioinformatics studies (*Rawlings et al., 2003*; *Withers et al., 2006*; *Higgs and Ran, 2008*; *Bermudez-Santana et al., 2010*; *Rogers et al., 2010*) and only a handful of experimental findings have been reported, which rely on genetic manipulations (*Byström and Fink, 1989*; *Von Pawel-Rammingen et al., 1992*; *Aström et al., 1993*) or direct mutagenesis (*Saks et al., 1998*). Sequence analyses of divergent genomes have demonstrated that both the sequence and copy number of tRNA genes may change among various species or strains. However, it is unclear whether the observed variations in the tRNA pool are a consequence of an adaptive process due to unbalanced translational demand or the result of random genomic processes, as tRNA genes are a known source of genomic instability (*McFarlane and Whitehall, 2009*).

Further, the forces that direct and maintain low copy tRNA families remain unclear. Specifically, it is not clear whether translational selection acts only to favor optimal codons or also acts in particular cases to keep other codons deliberately as 'non-optimal' by maintaining their tRNA supply at low level. Encoding genes with optimal codons might not always lead to higher protein expression levels (*Kudla et al., 2009*). Similarly, the use of 'slow codons' may not always result in lower levels of protein expression as they could have functional roles in improving expression, for example when enriched at the beginning of a transcript in order to reduce the energy of the RNA structure (*Goodman et al., 2013*) or to efficiently allocate ribosomes along the mRNA (*Tuller et al., 2010*). Additionally, it has been proposed that non-optimal codons may play a role in governing the process of cotranslational folding by slowing down translation, which supports proper folding between domain boundaries (*Thanaraj, 1996*; *Kramer et al., 2009*; *Cabrita et al., 2010*; *Wilke and Drummond, 2010*; *Pechmann and Frydman, 2012*). Yet, the contribution of non-optimal codons to proper protein folding was observed only for specific genes (*Crombie et al., 1992*; *Komar et al., 1999*; *Cortazzo et al., 2002*; *Tsai et al., 2008*; *Zhang et al., 2009*; *Zhou et al., 2013*). Thus, the extent and relevance of this phenomenon to the global folding state of the proteome needs to be substantiated.

To elucidate the importance of restoring translational equilibrium, we used an experimental evolution approach. To this end, we genetically perturbed the tRNA pool of the budding yeast *S. cerevisiae*. In this yeast, the genetic code is decoded by 42 different tRNA families that make up a total of 274 tRNA genes (*Chan and Lowe, 2009*). Each tRNA gene family ranges from 1–16 copies, with 6 tRNA families consisting of only a single copy. In a recent study (*Bloom-Ackermann et al., In press*), we have systematically manipulated the tRNA pool in *S. cerevisiae* by individually deleting most tRNA genes from its genome. Here, we focus on one particular deletion strain that showed the most extreme fitness reduction among the viable deletion mutants in this tRNA deletion library. This tRNA exists in only one copy in the genome, thus after its deletion the cell is left without a tRNA with the corresponding anticodon. Lab-evolution experiments performed on this strain demonstrated that the translational balance was rapidly restored by mutations in other tRNA genes that compensated for the tRNA deletion. An extensive bioinformatic analysis revealed that a similar evolutionary trend is widespread in nature too, suggesting that the anticodon mutations we observed in the lab recapitulate an existing mechanism that shapes the tRNA pool. To shed light on the constraints that shape the size of tRNA gene families, we artificially overexpressed singleton tRNAs, rather than deleting them. We found that when low copy tRNAs were overexpressed, the protein quality control machinery was challenged due to increased proteotoxic stress. This observation suggests that low tRNA availability for particular codon can serve an essential means to ensure proper cotranslation folding of proteins.

## Results

### Deletion of singleton tRNA gene breaks the translational balance

To demonstrate the importance of the balance between codon usage and the cellular tRNA pool we created a yeast strain in which the single copy of the arginine tRNA gene, *tR(CCU)J*, was deleted

(designated ΔtRNA$^{Arg}_{CCU}$). Consequently, in this deletion strain, the arginine codon AGG cannot be translated with its fully matched tRNA and it is presumably translated by another arginine tRNA, tRNA$^{Arg}_{UCU}$, owing to a wobble interaction (*Begley et al., 2007*). This shortage in tRNA supply is particularly evident given the demand: AGG is the second most highly used codon for arginine in the yeast genome (*Supplementary file 1A*). Indeed, the ΔtRNA$^{Arg}_{CCU}$ strain showed a severe growth defect compared to the wild-type strain (*Figure 1A*, blue and green curves, respectively). This growth difference demonstrates the effect of translational imbalance on cellular growth. Although the deletion mutant of this single copy tRNA is viable (*Clare et al., 1988*; *Kawakami et al., 1993*), its severe growth defect also reveals the inability of the wobble interactions to fully compensate for the tRNA gene deletion.

## The tRNA pool can rapidly evolve to meet translational demands

To learn how genomes adapt to translational imbalances, we performed lab-evolution experiments on the ΔtRNA$^{Arg}_{CCU}$ strain, employing the procedure of daily growth and dilution to a fresh medium (*Lenski et al., 1991*). The deletion strain was grown under optimal laboratory conditions (rich medium at 30°C) and was diluted daily into a fresh medium by a factor of 120, corresponding to approximately 7 generations per cycle. Every 50 generations, the growth of the evolving population was compared to both the wild-type and the ancestor ΔtRNA$^{Arg}_{CCU}$ strains. Strikingly, after 200 generations we observed a full recovery of the growth defect of the ancestor strain ΔtRNA$^{Arg}_{CCU}$, as the growth curve of the evolved population was indistinguishable from that of the wild-type strain (*Figure 1A*, red curve). Similar dynamics were observed in all four independent evolutionary lines.

In search of the potential genetic adaptations underlying this rapid recovery, we first looked for genetic alterations in other arginine tRNA genes. We found a single point mutation in another arginine tRNA gene that codes for tRNA$^{Arg}_{UCU}$. This mutation changed the anticodon triplet of tRNA$^{Arg}_{UCU}$ from UCU to CCU (i.e., T→C transition). Consequently, the evolved tRNA$^{Arg}_{UCU}$ perfectly matched the AGG codon (*Figure 1B*). Unlike the singleton tRNA$^{Arg}_{CCU}$, there are 11 copies of tRNA$^{Arg}_{UCU}$ in the yeast genome. Although each of the 4 independent lab-evolution experiments showed the exact same solution (that is, a mutation in the anticodon of a tRNA$^{Arg}_{UCU}$ gene), 3 different copies of this gene were changed among the 4 lines (i.e., 1 of the 11 copies of tRNA$^{Arg}_{UCU}$ was mutated in 2 repetitions; see 'Materials and methods'). To confirm that a single point mutation in the anticodon of tRNA$^{Arg}_{UCU}$ is sufficient to fully compensate for the growth defect of ΔtRNA$^{Arg}_{CCU}$, we artificially inserted the same T→C mutation into the deletion ΔtRNA$^{Arg}_{CCU}$ mutant. We inserted the mutation into 1 of the 11 copies of the tRNA$^{Arg}_{UCU}$ genes, a copy that resides on chromosome XI, and thus spontaneously mutated in 1 of the evolution lines. Indeed, the artificially mutated strain, termed as *MutΔtRNA$^{Arg}_{CCU}$*, showed a full recovery of the deletion adverse phenotype (*Figure 1C*). This indicates that the T→C mutation in the anticodon is sufficient for the full recovery of the tRNA$^{Arg}_{CCU}$ deletion phenotype.

## Mutated tRNA$^{Arg}_{UCU}$ is functional despite sequence dissimilarities with respect to the deleted tRNA$^{Arg}_{CCU}$

In general, all copies of each tRNA gene family tend to be highly similar in sequence in *S. cerevisiae* (*Chan and Lowe, 2009*). In particular, the sequences of the 11 copies of tRNA$^{Arg}_{UCU}$ are 100% identical to each other. Yet, the 2 arginine tRNA, tRNA$^{Arg}_{UCU}$, and tRNA$^{Arg}_{CCU}$, differ in 21 of their 72 nucleotides (including the third anticodon position, *Figure 2A*). Thus, the evolutionary solution that occurred in our experiments created a 'chimeric' tRNA with a CCU anticodon, whereas the rest of the tRNA sequence (termed as the 'tRNA scaffold') remained as tRNA$^{Arg}_{UCU}$. The sequence identity among all members of the tRNA$^{Arg}_{UCU}$ family suggests a functional role for the tRNA scaffold in addition to that of the anticodon (*Schultz and Yarus, 1994*; *Konevega et al., 2004*; *Cochella and Green, 2005*; *Olejniczak et al., 2005*; *Saks and Conery, 2007*; *Schmeing et al., 2011*). Therefore, it is surprising that the chimeric tRNA performed just as well as the deleted tRNA$^{Arg}_{CCU}$ in terms of cell growth, despite the major sequence differences between the two tRNA scaffolds. Thus, we raised the hypothesis that more challenging growth conditions may expose possible inadequacies in the chimeric tRNA. To test this notion, we compared the rescued strain, *MutΔtRNA$^{Arg}_{CCU}$*, which carries the chimeric tRNA, to the wild-type strain under an array of challenging conditions. Surprisingly, under all the examined conditions, we observed no significant growth difference between the two strains (*Figure 2B*). Hence, the chimeric tRNA provides a direct in vivo indication that the scaffolds of tRNAs, which encode for the same amino acid, may be interchangeable in terms of their effect on cellular growth under the conditions we tested.

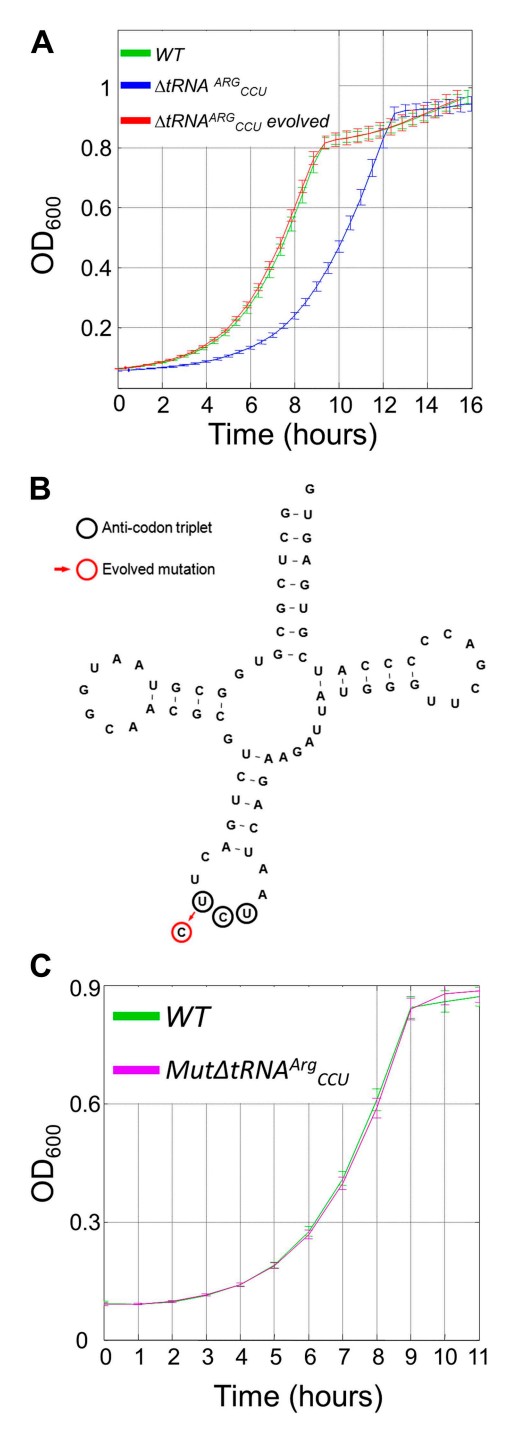

**Figure 1**. The growth defect associated with deletion of a singleton tRNA gene was rapidly rescued during the lab-evolution experiment. (**A**) Growth curve measurements of wild-type (WT) (green), ΔtRNA$^{Arg}_{CCU}$ (blue) and the evolved deletion (red) are shown in optical density (OD) values over time during continuous growth on rich medium at 30°C. (**B**) The mutation that was found to recover the deletion phenotype in the evolved strains is shown on the secondary structure of tRNA$^{Arg}_{UCU.}$

*Figure 1. Continued on next page*

To examine the generality of our observation, we once again perturbed the tRNA pool in a wild-type (WT) strain by deletion of an entire serine tRNA family, tRNA$^{Ser}_{GCU}$ that has four copies in the genome. A complete deletion of this gene family is lethal, indicating that the tRNA$^{Ser}_{GCU}$ is essential in *S. cerevisiae*. Although we could not evolve that strain, we did find that this quadruple deletion strain was viable when supplemented with a centromeric plasmid carrying the tRNA$^{Ser}_{GCU}$ gene. Thus, the lethality is conferred directly from the tRNA loss and is not due to other indirect effects (*Figure 2—figure supplement 1*). We hypothesized that, as with tRNA$^{Arg}_{CCU}$, other chimeric serine tRNAs that carry a GCU anticodon, yet with the scaffold of another tRNA for serine, would also prevent the observed lethality. Indeed, a strain carrying a chimeric tRNA with a scaffold of tRNA$^{Ser}_{CGA}$ and the GCU anticodon is viable on the background of the tRNA$^{Ser}_{GCU}$-family deletion (*Figure 2—figure supplement 2*). Therefore, we concluded that the identity of the anticodon is essential for the function of the tRNA$^{Ser}_{GCU}$ gene family. Thus, it appears that for the examined tRNAs the anticodon is a dominant feature in terms of cellular fitness, overshadowing other sequence elements.

## Anticodon switching is a widespread phenomenon in nature

Although the anticodon of a tRNA gene was rapidly mutated under our laboratory conditions, thus regaining proper translational equilibrium, it is unclear to what extent this mechanism naturally occurs in species across the tree of life. To address this question, we performed a systematic bioinformatics screen for tRNA switching events in nature. We defined an anticodon-switching event as a case of a tRNA whose nucleotide sequence is closer to a tRNA gene with a different anticodon than to a tRNA gene with the same anticodon. To this end, we downloaded all the known tRNA sequences from the Genomic tRNA Database (*Chan and Lowe, 2009*), a collection that stores the tRNA pools of 524 species. We masked the anticodon triplet as 'NNN' in all tRNA genes, aligned all tRNA sequences from each species individually and inferred a maximum likelihood phylogenetic tree for each alignment. For each tRNA sequence, we calculated the shortest phylogenetic distance to another tRNA with the same anticodon (designated $d_{same}$) and the shortest distance to another tRNA with a different anticodon (designated $d_{diff}$). For each species, we defined its set of tRNA switching events as those in which $d_{diff} < d_{same}$ (see 'Materials and methods', *Figure 3—source data 1*).

*Figure 1. Continued*

The UCU anticodon nucleotides are marked with black circles, and the red circle indicates the mutation that occurred in the anticodon, that is T→C transition. (**C**) *MutΔtRNA$^{Arg}_{CCU}$* in which the same mutation that was found in the evolved strain was deliberately engineered, exhibits similar growth as the WT. Growth curve measurements of WT (green) and of *MutΔtRNA$^{Arg}_{CCU}$* (magenta) are shown in OD$_{600}$ values over time during continuous growth on rich medium at 30°C.

Our analysis included 416 eubacterial, 68 eukaryotic, and 40 archaeal species. We found that tRNA switching events are present in all domains of life, as we detected at least 1 tRNA switching event per species in 8 bacteria, 58 eukarya, and 1 archaeal species (*Figure 3A*). A retrospective counting revealed that most switching events occurred due to a mutation in the first position in the anticodon triplet that corresponds to the third codon position (see details in 'Materials and methods'). For comparison, we masked as 'NNN' additional triplets of nucleotides within the tRNA molecule, and found a higher percentage of discrepancies compared to the anticodon triplet (*Figure 3—figure supplement 1*; *Supplementary file 1B*).

*Figure 3* demonstrates two examples of tRNA switching events, the first in *Mus musculus* and the second in *Homo sapiens*. In the first example, the phylogeny of tRNA sequences with glutamic acid anticodons is presented (*Figure 3B*). Notably, six out of the eight tRNAs with a UUC anticodon in *M. musculus* were clustered together in our analysis, while two other copies of the same anticodon identity were clustered closer to tRNA genes with a CUC anticodon (*Figure 3C*). The second example demonstrates a switching event for tRNA genes encoding for valine anticodons. In this study, a tRNA with a UAC anticodon was clustered with CAC and AAC tRNA genes and not with the other four UAC tRNAs (*Figure 3D,E*). Interestingly, the CAC and AAC tRNA genes are intermixed in the tree, suggesting that anticodon switching was prevalent in the evolution of CAC and AAC tRNA genes in *H. sapiens* (*Figure 3D,E*). Also of interest, the switching events shown in mouse were not found in human and vice versa. Thus, in each of these two mammals the switching examples shown here probably occurred after they split from their common ancestor. In general, inspecting the relationship across species between the size of the tRNA pool and the number of detected switching events revealed a modest correlation, and in particular species with same size of tRNA repertoire manifested tRNA switching to different extents (not shown). This analysis suggests that future examination of the tRNA switching phenomenon in individual species could be of interest.

## Multiple copies of rare tRNAs are deleterious to the cell

After demonstrating the prevalence of anticodon switching, we refocused on our lab-evolution results. The switching events that we observed (from tRNA$^{Arg}_{UCU}$ to tRNA$^{Arg}_{CCU}$) suggest that the effective gene copy number of each tRNA anticodon set can change during evolution, presumably due to demand-to-supply changes. Given that a single point mutation can functionally convert one tRNA into another, an interesting question emerges: why does the genome maintain a single copy of tRNA$^{Arg}_{CCU}$? T to C mutations must have occurred in evolution but they appear to have been selected against so as to preserve only a single copy of the CCU anticodon tRNA. Consistent with this hypothesis is the observation that other yeast species maintain tRNA$^{Arg}_{CCU}$ at a single copy (*Supplementary file 1C*). We thus reasoned that an artificial increase in the copy number of a rare tRNA, but not of an abundant one, might result in a deleterious effect on the cells.

Indeed, transformation of a multi-copy plasmid containing a tRNA$^{Arg}_{CCU}$ gene to a wild-type strain (termed as *WTmultiCCU*) resulted in a substantial growth reduction when compared to wild-type cells carrying an empty multi-copy plasmid (termed *WTmultiControl*). In contrast, when we created a strain with a similar multi-copy plasmid that contains the abundant tRNA$^{Arg}_{UCU}$ gene, designated as *WTmultiUCU*, a growth profile much closer to that of *WTmultiControl* was observed (*Figure 4A*). These findings are consistent with the evolutionary tendency for yeast to keep a low copy number of tRNA$^{Arg}_{CCU}$ and suggest that a high copy number of such rare tRNAs is deleterious to cells.

To demonstrate the generality of our findings, we employed the same assays in two additional cases. First, we examined 2 serine tRNAs, the singleton tRNA$^{Ser}_{CGA}$ and tRNA$^{Ser}_{AGA}$ that is found in the genome in 11 copies. In the second case, we focused on two glutamine tRNAs, the singleton tRNA$^{Gln}_{CUG}$ and tRNA$^{Gln}_{UUG}$ that is found in the genome in nine copies. In both the cases, we observed that the wild-type strain supplemented with multiple copies of a singleton tRNA exhibit impaired growth compared to the same strain supplemented with the abundant tRNA for the same amino acid (*Figure 4—figure supplements 1 and 2*). Since the changes in tRNA family sizes during evolution likely occur gradually,

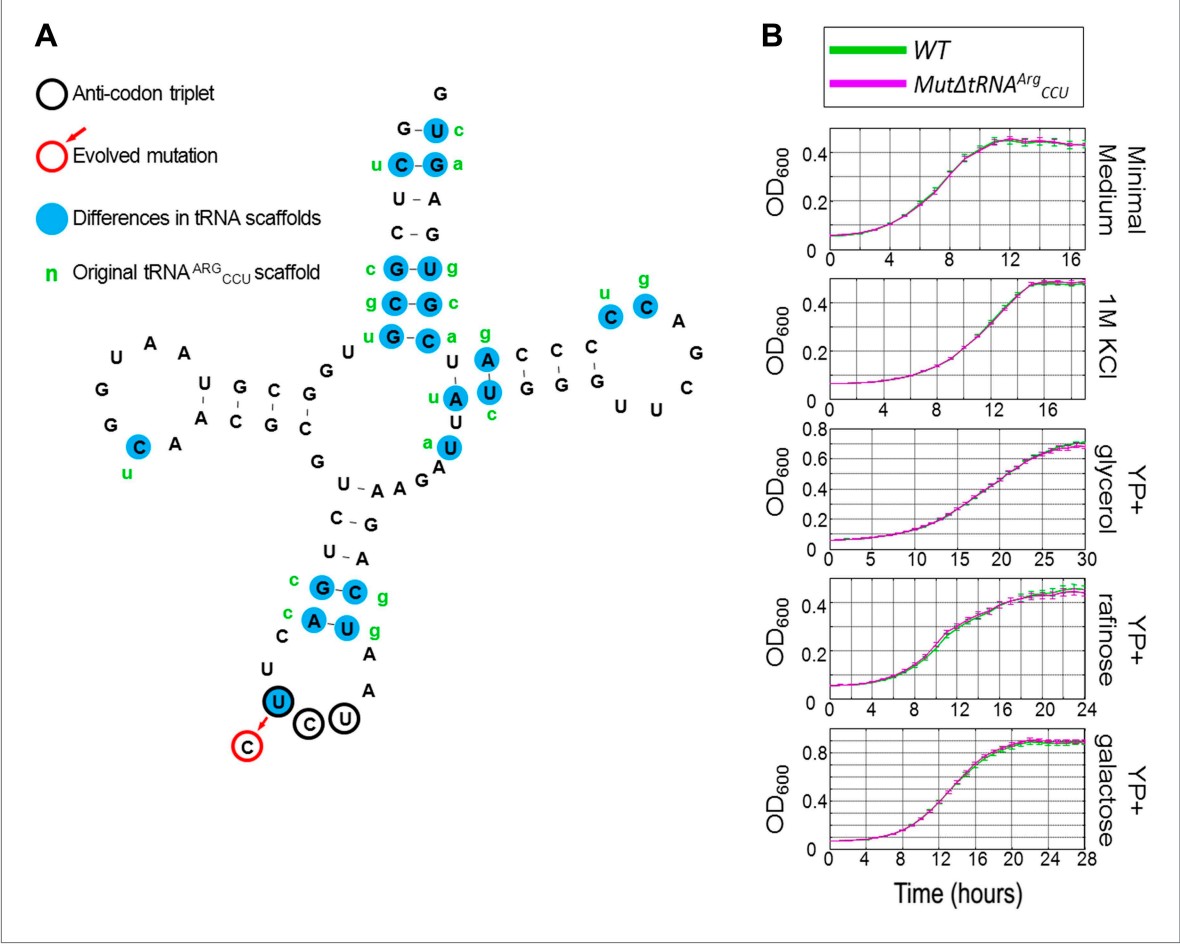

**Figure 2**. The growth of MutΔtRNA$^{Arg}_{CCU}$ carrying the chimeric tRNA compared to wild-type (WT) under different conditions. (**A**) The sequence of the chimeric tRNA is drawn showing the scaffold of tRNA$^{Arg}_{UCU}$ with the mutated CCU anticodon. The anticodon triplet is marked with black circles. The evolved mutation is marked with a red circle. All 20 nucleotide differences between tRNA$^{Arg}_{UCU}$ and tRNA$^{Arg}_{CCU}$ are marked with blue background, next to which, in green letters, the original nucleotide of tRNA$^{Arg}_{CCU}$ are written. (**B**) Growth curve measurements of WT (green) and of *MutΔtRNA$^{Arg}_{CCU}$* (magenta) are shown in OD$_{600}$ values over time during continuous growth.

The following figure supplements are available for figure 2:

**Figure supplement 1**. Quadruple deletion of tRNA$^{ser}_{GCU}$ is lethal.

**Figure supplement 2**. A chimeric serine tRNA can rescue the lethality of the quadruple deletion.

perhaps one copy at a time, we also examined the effect of adding low copy number plasmids carrying either tRNA$^{Arg}_{UCU}$ or tRNA$^{Arg}_{CCU}$. The cells with the tRNA$^{Arg}_{CCU}$ plasmid showed a modest growth defect compared to the cells with tRNA$^{Arg}_{UCU}$ plasmid, yet only when grown at 39°C (*Figure 4—figure supplement 3*).

## Multiple copies of the rare tRNA$^{Arg}_{CCU}$ induce proteotoxic stress

Why is it essential to keep certain tRNAs at a low level? One interesting possibility is that rare tRNAs are essential for the process of cotranslation folding, presumably because low abundance tRNAs provide a pause in translation that might be needed for proper folding (*Thanaraj, 1996*; *Drummond and Wilke, 2008*; *Cabrita et al., 2010*; *Pechmann and Frydman, 2012*). Other deleterious effects that may stem from a high copy number of tRNA$^{Arg}_{CCU}$ could be misincorporation of arginine into non-arginine codons, or the misloading of arginine tRNA molecules with other amino acids. These potential sources of errors are not mutually exclusive and can each contribute to the observed growth defect by exerting a protein folding stress.

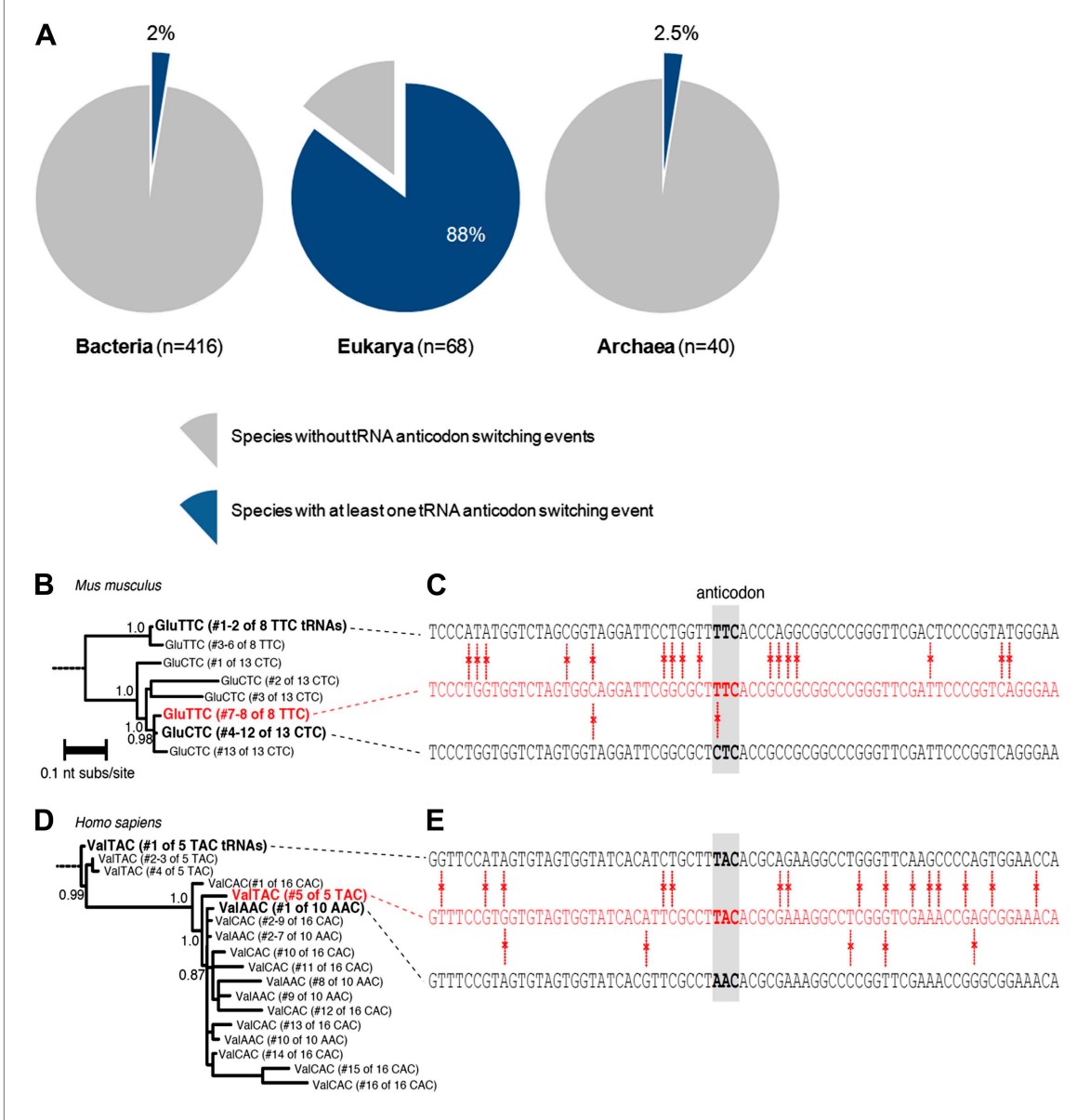

**Figure 3**. Anticodon switching is a widespread phenomenon in nature. (**A**) Number of species with at least one tRNA switching event in each domain of life. (**B**) The anticodon UUC convergently evolved in *Mus musculus*. A maximum likelihood phylogeny of tRNA sequences in *M. musculus* that decode glutamic acid (Glu) codons. Branch lengths express average nucleotide substitutions per site. Decimals on internal branches express branch support. (**C**) A comparison of nucleotide sequences for glutamic acid tRNA genes in *M. musculus* with anticodon UUC (top, tRNA1547 and tRNA359), 'switched' UUC tRNAs (middle, tRNA286 and tRNA754), and CUC tRNAs (bottom, tRNA1002, tRNA745, tRNA303, tRNA999, tRNA996 tRNA709, tRNA1001, tRNA1912 and tRNA81). The anticodon triplet is boxed in gray. Red vertical bars indicate differences between sequences. (**D**) The anticodon UAC convergently evolved in *Homo sapiens*. A maximum likelihood phylogeny of tRNA sequences in *H. sapiens* encoding for valine (Val) is shown. (**E**) A comparison of nucleotide sequences for *H. sapiens* tRNAs with anticodons UAC (top, tRNA6), a 'switched' UAC tRNA (middle, tRNA40), and an AAC tRNA (bottom, tRNA136). The number of genes is according to the tRNA database.

The following source data and figure supplements are available for figure 3:

**Source data 1**. Table of anticodon switchings in different species across the tree of life.

**Figure supplement 1**. A comparison of discrepancy proportions at the anticodon triplet vs control triplets.

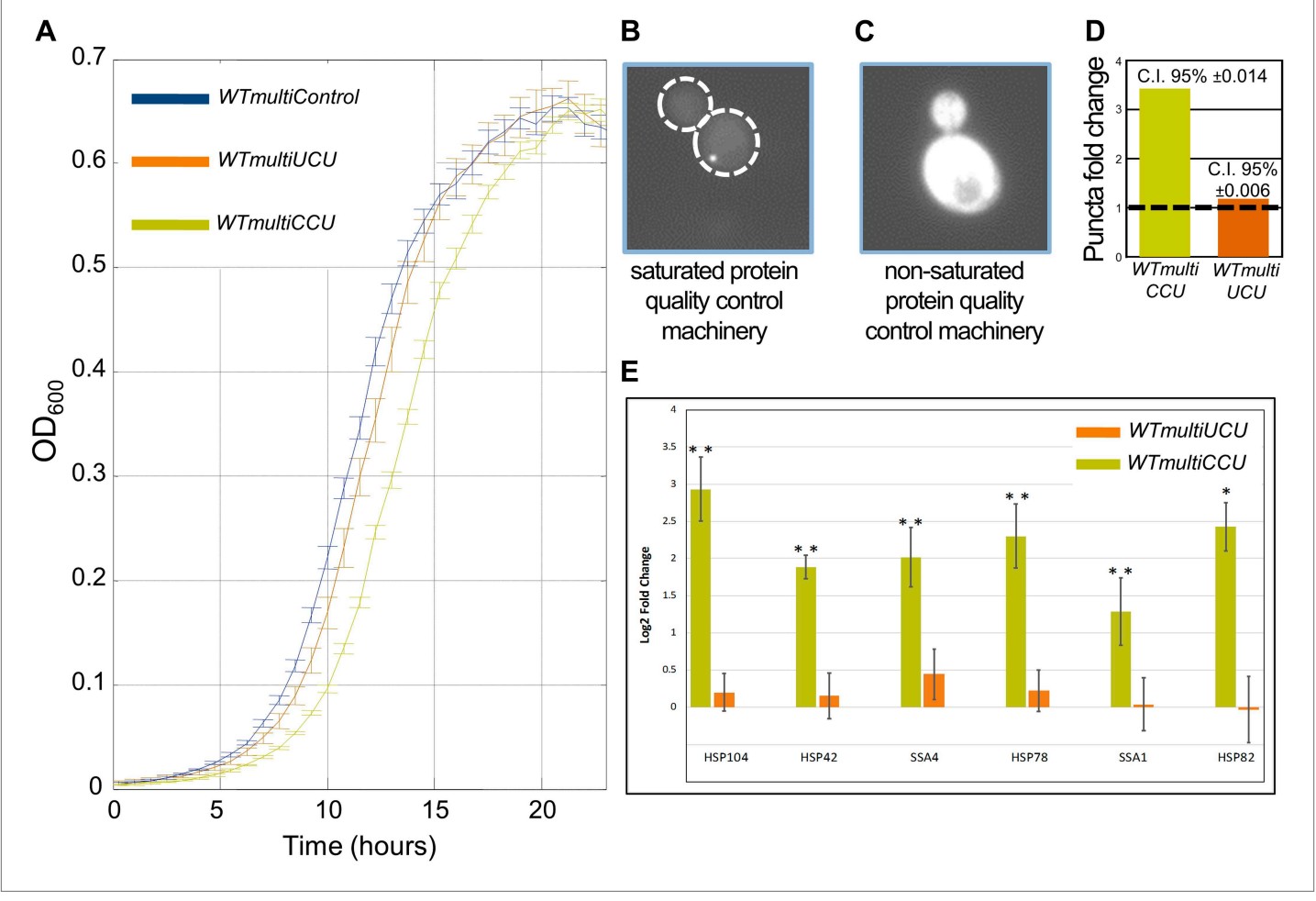

**Figure 4**. *WTmultiCCU* experiences a growth defect compared to *WTmultiUCU* and demonstrates higher levels of misfolded proteins. (**A**) Growth curve measurements of *WTmultiControl* (blue), *WTmultiUCU* (brown) and *WTmultiCCU* (khaki) are shown in optical density (OD) values over time during continuous growth. The *WTmultiCCU* strain carrying a high copy number plasmid harboring tRNA$^{Arg}_{CCU}$ demonstrates slower growth compared to cells with an empty plasmid or with a tRNA$^{Arg}_{UCU}$ plasmid that is mainly characterized by a longer growth delay (lag phase). (**B**) A demonstration of a *WTmultiCCU* cell in which the mCherry-Von Hippel–Lindau (VHL) proteins appear with a punctum phenotype when the protein quality control machinery is saturated with misfolded proteins. (**C**) A demonstration *WTmultiUCU* cell in which the quality control machinery is not occupied with other proteins; mCherry-VHL is localized to the cytosol. (**D**) *WTmultiCCU*, *WTmultiUCU* and *WTmultiControl* were transformed with a VHL-mCherry containing plasmid and visualized under the microscope; 1000 cells per strain were counted for either cytosolic or punctum localization of the VHL protein. The fold change in the number of cells containing puncta was then deduced by normalization to the *WTmultiControl* population. The 95% confidence interval is indicated. (**E**) The mRNA fold change of six representative heat-shock genes measured by real-time quantitative PCR (RT-qPCR). Presented values are the mean of two biological repetitions ± SEM. The significance of the fold change differences was examined using a *t* test, with *p<0.001 or **p<0.0001.

The following figure supplements are available for figure 4:

**Figure supplement 1**. Multiple copies of rare tRNA$^{Ser}_{CGA}$ gene are deleterious compared to abundant tRNA$^{Ser}_{AGA}$.

**Figure supplement 2**. Multiple copies of the rare tRNA$^{Gln}_{CUG}$ gene are deleterious compared to abundant tRNA$^{Gln}_{UUG}$.

**Figure supplement 3**. Addition of low copy number tRNA$^{Arg}_{CCU}$ is deleterious compared to low copy number tRNA$^{Arg}_{UCU}$ when grown in heat.

To examine the possibility that the growth defect associated with multiple copies of tRNA$^{Arg}_{CCU}$ is indeed associated with such a proteotoxic stress, we used an established method that examines the load on the protein quality control machinery of the cell (see 'Materials and methods') (*Kaganovich et al., 2008*). In this assay, we transformed cells with a plasmid harboring the human gene, von Hippel–Lindau (VHL), fused to a fluorescent tag (mCherry). Fluorescently tagged VHL that is present as

aggregated puncta (*Figure 4B*), and not as a disperse cytosolic localization (*Figure 4C*), indicates that the protein quality control machinery is saturated due to high levels of misfolding in the cell's endogenous proteins. We transformed the VHL-mCherry plasmid to each of the multi-copy tRNA strains, *WTmultiCCU*, *WTmultiUCU* and *WTmultiControl,* and monitored the level of proteotoxic stress by quantifying the number of cells with puncta phenotype in each population. The fold change in those cells was then deduced by normalization to the *WTmultiControl* population. We found that while *WTmultiUCU* exhibited similar amount of cells with puncta as the *WTmultiControl*, the *WTmultiCCU* exhibited a threefold increase (*Figure 4D*).

The proteotoxic stress experienced by the two strains overexpressing tRNAs was further assessed by measuring the induction level of an array of heat-shock proteins (HSPs) using real-time quantitative PCR (RT-qPCR). Since the HSPs have been shown to undergo induction under proteotoxic stress, they are an excellent indicator for this stress (*McClellan et al., 2005*; *Kaganovich et al., 2008*). Indeed, we found a significant upregulation in mRNA levels for all the examined HSP genes in the *WTmultiCCU* strain, but not in the *WTmultiUCU* strain (*Figure 4E*). These findings further demonstrate that increasing the copy number of a rare tRNA gene, but not of an already abundant one, results in proteotoxic stress in the cell.

## Discussion

Genomic duplications, deletions, and anticodon mutations shape tRNA gene families, yet the evolutionary scenarios that trigger changes in the tRNA pool have not been thoroughly explored. In our evolution experiments, a translational imbalance was imposed by a tRNA gene deletion that compromised growth and drove the tRNA pool to adapt to a novel translational demand. Importantly, organisms may experience equivalent imbalances when their gene expression changes due to altered environmental conditions or upon migrating to a new ecological niche (*Gingold et al., 2012*). This scenario is particularly feasible given that the genes needed in various environments do show differences in codon usage, for example respiration as opposed to fermentation in yeast (*Man and Pilpel, 2007*).

Indeed, when faced with different environmental challenges, transcriptional changes affect the codon usage of the transcriptome (*Gingold et al., 2012*), and hence the demand for the various tRNAs, and may thus cause translational imbalances. To maintain optimal protein production, the tRNA pool is under pressure to restore the translational balance by accommodating the new translational demands. On a short timescale, the tRNA pool might respond non-genetically by changing expression profiles of the tRNAs (*Tuller et al., 2010*; *Saikia et al., 2012*; *Pavon-Eternod et al., 2013a*, *2013b*). Yet, if changes in demand-to-supply persist, a genetic change in the tRNA pool might become beneficial evolutionarily. In this work, we demonstrate how anticodon mutations provide a rapid mechanism to alter the tRNA pool. We propose that during evolution, novel translational requirements can be addressed by anticodon shifting of tRNA copies more readily than by duplications and deletions of tRNA genes. The tRNA pool can evolve to meet new translation demands by adjusting the ratios of tRNA families that code for the same amino acid. Within a single mutational event, anticodon switching holds the potential to rapidly change the ratios of tRNAs within the pool, by increasing the copy number of one tRNA family at the expense of a counterpart. A similar solution could be obtained by a sequence of genomic duplications and deletions of tRNA genes. These alternatives are likely to fixate less frequently than anticodon switching, as they may carry negative effects due to duplications or deletions of adjacent unrelated genetic features. Furthermore, our systematic search for tRNA switching events throughout the tree of life revealed the prevalence of tRNA anticodon mutations in nature. This observation is consistent with the results of our lab-evolution experiment and may be the evolutionary outcome to novel translational demands in the wild.

Studies on methionine tRNAs have previously shown that the scaffold sequences determine their function as either initiator or elongator Met tRNA (*Von Pawel-Rammingen et al., 1992*; *Aström et al., 1993*; *Kolitz and Lorsch, 2010*). Yet, the initiator and the elongator represent extreme cases of tRNAs that are used at different stages in translation. The present chimeric tRNAs that emerged in our lab-evolution experiments successfully replaced the deleted tRNA, despite differences in 20 nucleotide positions between the 2 tRNA scaffolds. If tRNA scaffolds are interchangeable in terms of the effect of their function on the fitness, what can explain the high sequence similarity observed among tRNA gene copies of the same family in yeast? It is possible that the sequence of the tRNA scaffold is indeed important under specific conditions that were not examined in this work, or that our measurements

were not sensitive enough to detect small selective disadvantages that can act against chimeric tRNAs in nature. Under these scenarios, the high sequence similarity can be explained by purifying selection that maintains sequence identity within tRNA families. Yet, there is also a possibility that the sequence similarity is not due to purifying selection but the result of 'concerted evolution', an evolutionary process that maintains sequence identity by frequent recombination events among copies of the same gene family (*Munz et al., 1982*; *Amstutz et al., 1985*; *Teshima and Innan, 2004*). This possibility implies that the high conservation observed within tRNA gene families is not due to functionality, but is rather the result of neutral evolution. At present, it is not possible to determine which of the two possibilities explains best the observed sequence identity.

If a single point mutation in one of the tRNA$^{Arg}_{UCU}$ copies enables it to function like a tRNA from a different family, what were the evolutionary constraints that have left some families with more members while others with fewer? Is purifying selection acting to deliberately maintain low levels of certain tRNAs? Such selection would render their corresponding codons 'non-optimal'. To examine potential adaptive functions of tRNA family sizes, we tested the consequences of increasing the sizes of several tRNA families. We found that keeping low copy tRNA families is adaptive, as increasing their copy number can result in a proteotoxic stress due to problems in protein folding.

Most of the published work on the functionality of codons that correspond to rare tRNAs have so far tested how modified codon usage of specific proteins influences their proper folding (*Crombie et al., 1992*; *Komar et al., 1999*; *Cortazzo et al., 2002*; *Tsai et al., 2008*; *Zhang et al., 2009*; *Zhou et al., 2013*). In contrast, we took a different approach, in which no protein coding gene sequence is modified, but rather the tRNA supply is manipulated. Thus, the effect we generated could be exerted on all genes, and we could indeed detect it as a global proteotoxic stress in the cell. Our observations are consistent with the theory that programmed pauses in the translation process could promote proper folding during translation (*Thanaraj, 1996*; *Kramer et al., 2009*; *Cabrita et al., 2010*; *Wilke and Drummond, 2010*). The overexpression of the rare tRNA could have thus impaired with cotranslation folding. Yet, there could be additional reasons for the observed proteotoxic stress, which are not necessarily mutually exclusive. First, overexpression of tRNA$^{Arg}_{CCU}$ may result in misincorporation of arginine into non-arginine codons. Second, other aminoacyl tRNA synthetases may aminoacylate an incorrect amino acid to the highly expressed tRNA. Misloading will result in the incorporation of a different amino acid instead of arginine. A potential part of the observed proteotoxic stress due to misincorporation still remains to be studied. Yet, such an effect should be relevant not only for the overexpression of the rare tRNA$^{Arg}_{CCU}$ but also for the overexpression of the abundant tRNA$^{Arg}_{UCU}$. Our results show a sever proteotoxic stress only upon expression of the rare tRNA, thus landing more support to the intriguing hypothesis, that the proteotoxic phenotypes observed are due to converting a slow translating codon, scattered in many genes in the genome, into a fast one. This notion is consistent with and complementary to the picture that emerges from single gene-based analyses.

When facing the need to adapt, the tRNA pool (that is the supply) provides evolutionary plasticity to the translation machinery. The ability of the tRNA pool to change rapidly can be mainly attributed to its unique architecture in the form of multimember gene families. Only on a much longer evolutionary timescale, will the genome-wide codon usage of genes change so as to further fine tune the translational balance. Notably, the plasticity of the tRNA genes is constrained by the need to maintain proper protein folding (*Drummond and Wilke, 2008*). Thus, the need to accommodate changes in codon usage demands acts together with protein folding constrains to shape the tRNA pool in the living cells.

## Materials and methods

### Yeast strains and plasmids

The following *S. cerevisiae* strains were used in this study: ΔtRNA$^{Arg}_{CCU}$ (based on Y5565, genetic background: *ΔtR(CCU)J::Hyg, MATα, can1Δ::MFA1pr-HIS3 mfα1Δ::MFα1pr-LEU2 lyp1Δ ura3Δ0 leu2Δ0*) (*Bloom-Ackermann et al., In press*) was used for lab-evolution experiments. *MutΔtRNA$^{Arg}_{CCU}$* is based on ΔtRNA$^{Arg}_{CCU}$ and carries a mutation (T→C transition) in *tR(UCU)K* gene plus a *URA3* selection marker. BY384 (*MATa leu2Δ1 lys2Δ202 trp1Δ63 ura3-52 his3Δ200*) was used to generate a complete deletion of the tRNA$^{Ser}_{GCU}$ gene family. BY4741 (*MATa his3Δ1 leu2Δ0 met15Δ0 ura3Δ0*) and BY4742 (*MATα his3Δ1 leu2Δ0 lys2Δ0 ura3Δ0*) were used for examining the effect of increasing tRNA gene copy number.

Plasmids used in this study to express tRNA genes were pRS316 (CEN, *URA3*), pRS425 (2μ, *LEU2*), and pRS426 (2μ, *URA3*). For the rescue assays of the quadruple serine deletion, the pQF50 (2μ, *URA3*) and pQF150 (2μ, *LEU2*) were used. For the protein quality control assays the pGAL-VHL-mCherry (2μ, *LEU2*) plasmid was used (*Kaganovich et al., 2008*). For additional information on plasmids and primers see *Supplementary file 2*.

## Media

Cultures were grown at 30°C in either rich medium (1% bacto-yeast extract, 2% bacto-peptone and 2% dextrose [YPD]) or synthetic medium (0.67% yeast nitrogen base with ammonium sulfate and without amino acids and 2% dextrose, containing the appropriate supplements for plasmid selection). Protein quality control assays were performed on synthetic medium supplemented with 2% galactose as a carbon source. All chemicals used to create the media were manufactured by BD. All sugars, nucleic acids and amino acids were manufactured by Sigma-Aldrich.

## Evolution experiments

Lab-evolution experiments were carried out by serial dilution. Cells were grown on 1.2 ml of YPD at 30°C until reaching stationary phase and then diluted by a factor of 1:120 into fresh media (6.9 generations per dilution). This procedure was repeated daily until population growth under the applied condition matched the wild type. In all measurements of evolved populations, we used a population sample and not selected clones.

## Liquid growth measurements

The cultures were grown at the relevant condition, and optical density (OD)$_{600}$ measurements were taken during the growth at 30–45 min intervals until reaching early stationary phase. Qualitative growth comparisons were performed using 96-well plates (Thermo Scientific) in which 2 strains were divided on the plate in a checkerboard manner on the plate to cancel out positional geographical effects. For each strain, a growth curve was obtained by averaging over 48 wells.

## Growth on 5-fluoro-orotic acid (5-FOA) plates

Strains were grown for 2 days in a non-selective liquid medium, which contains uracil (YPD), to allow growth of cells that lost the plasmid containing the *URA3* counterselectable marker (*Boeke et al., 1984*). Then, 100 μl were plated on a YPD plate and replicated on the following day on either YPD or standard 5-fluoro-orotic acid (5-FOA) (US Biological) plates to identify potential colonies that lost the plasmid. Following 2 days of incubation at 30°C, growth of the colonies was scored.

## Measurements for saturation of the protein quality control machinery

We used a previously published method that allows examination of the protein quality control of the cell (*Kaganovich et al., 2008*). This assay provides an indication for the protein unfolding stress in cells by assessing the load on the protein quality control machinery. In this assay, the cells were introduced with a high copy number plasmid that contains the human gene VHL fused to a fluorescent tag (mCherry). VHL is a naturally unstructured protein and is dependent on two additional proteins (Elongin B and C) for proper folding in human cells. Expressing VHL in yeast cells, which lack VHL's complex partners, leads to misfolding of the translated proteins. Under normal conditions, the misfolded VHL proteins are handled by the cell's quality control machinery. When the quality control machinery is not saturated, the fluorescently tagged VHL appears in the cytosol. However, under stress, in which the quality control machinery is fully occupied, misfolded proteins in the cytosol are processed into dedicated inclusions (JUNQ and IPOD) and form punctum structures. Hence, a punctum phenotype of the VHL-mCherry construct is an indication for cells that experience proteotoxic stress and saturation of the protein quality control machinery.

Wild-type yeast cells harboring the pGAL-VHL-mCherry (CHFP) fusion plasmid (*Kaganovich et al., 2008*) and either an additional empty plasmid or the tRNA overexpression plasmid, were grown overnight on SC+2% raffinose, diluted into SC+2% galactose and grown at 30°C for 6 hr. The cells were visualized using an Olympus IX71 microscope (Olympus) controlled by Delta Vision SoftWoRx 3.5.1 software, with X60 oil lens. Images were captured by a Photometrics Coolsnap HQ camera with excitation at 555/28 nm and emission at 617/73 nm (mCherry). The images were scored using ImageJ image processing and analysis software. The percentage of cells harboring VHL-CHFP foci (Puncta) in the overexpression strains were normalized to the level in a control strain carrying an empty plasmid.

## Computational identification of anticodon switching events

To characterize the extent of anticodon switching across the tree of life, we first downloaded all tRNA sequences from the Genomic tRNA Database (*Chan and Lowe, 2009*). The sequences in this database were discovered with the tRNAscan algorithm (*Lowe and Eddy, 1997*), which finds tRNA sequences by scanning genomic DNA. We removed psuedogene tRNAs, which are defined as those tRNAs with a COVE score less than 40.0 (*Lowe and Eddy, 1997*) . For each remaining tRNA sequence, we masked its anticodon triplet as 'NNN'. We next grouped all tRNA sequences by their species, and then aligned the sequences for each species using Muscle with default settings (*Edgar, 2004*). For each species, we inferred a maximum likelihood phylogeny of its tRNA sequences using RAxML with the GTRCAT model (*Stamatakis, 2006*). We calculated statistical support for tree branches using SH-like approximate likelihood ratio test (*Anisimova and Gascuel, 2006*). We next interrogated each species' tRNA phylogeny using DendroPy (*Sukumaran and Holder, 2010*). Specifically, we identified those tRNA sequences harboring an anticodon that appears in the genome more than once; for each of these tRNA sequences, we found the shortest distance to another tRNA with the same anticodon ($d_{same}$) and the shortest distance to another tRNA with a different anticodon ($d_{diff}$). We labeled tRNAs as putatively 'switched' if $d_{diff} < d_{same}$.

## RT-qPCR measurements of HSP genes in strains overexpressing tRNAs

Cultures were grown in rich medium at 30°C to a cell concentration of $1 \times 10^7$ cells/ml. Then, RNA was extracted using MasterPure kit (Epicentre-illumina) (EPICENTER Biotechnologies), and used as a template for quantitative RT-PCR using light cycler 480 SYBR I master kit (Roche Applied Science) and the LightCycler 480 system (Roche Applied Science), according to the manufacturer's instructions.

## Genomic copies of tRNA$^{Arg}_{UCU}$ mutated during lab-evolution experiments

In all, 4 independent lab-evolution experiments that started with $\Delta$tRNA$^{Arg}_{CCU}$ as the ancestral strain showed full recovery of the deletion phenotype after 200 generations. In each of the evolved populations a mutation in one of the copies of tRNA$^{Arg}_{UCU}$ was found to change the anticodon from UCU to CCU. The genomic copies of tRNA$^{Arg}_{UCU}$ that were found to carry the mutation were: *tR(UCU)K*, *tR(UCU) G1* and *tR(UCU)D* that was changed in two of the independent cultures.

## The contribution of different anticodon positions to tRNA switching events

Out of 4245 anticodon switching events that we detected, the first position in the anticodon was changed in 2540 cases while the second and third were only involved in 1448 and 1330 cases, respectively.

# Acknowledgements

We thank M Schuldiner, D Kaganovich, and S Leidel for kindly providing plasmids. We also thank the Pilpel lab, and especially H Gingold, for fruitful discussions. We acknowledge T Ast, Y Cohen and R Ackermann for critical reading of the manuscript. We thank the European Research Council (ERC) (YP), the Ben-May Charitable Trust (YP) and the NIH (grant 1P50GM107632) (JDB) for grant support. YP is an incumbent of the Ben May Professorial Chair.

# Additional information

### Funding

| Funder | Grant reference number | Author |
| --- | --- | --- |
| European Research Council | ERC-2007-StG 205199-ERNBPTC | Yitzhak Pilpel |
| Ben-May Charitable Trust | | Yitzhak Pilpel |

The funders had no role in study design, data collection and interpretation, or the decision to submit the work for publication.

### Author contributions

AHY, IF, Conception and design, Acquisition of data, Analysis and interpretation of data, Drafting or revising the article; ZB-A, Design, Acquisition of data, Analysis and interpretation of data, Drafting or

revising the article; VH-S, Acquisition of data, Analysis and interpretation of data, Drafting or revising the article; YC-A, Acquisition of data; QF, Acquisition of data, Drafting or revising the article, Contributed unpublished essential data or reagents; JDB, Analysis and interpretation of data, Drafting or revising the article, Contributed unpublished essential data or reagents; OD, Design, Analysis and interpretation of data, Drafting or revising the article; YP, Conception and design, Analysis and interpretation of data, Drafting or revising the article

## Additional files

### Supplementary files

• Supplementary file 1. Usage of arginine codons in *Saccharomyces cerevisiae*. (**A**) The AGG codon constitutes approximately 21% of the arginine codons in the yeast genome and approximately 16% of the arginine codons in the yeast transcriptome under standard lab conditions. In comparison, the AGA codon constitutes approximately 47.5% of the arginine codons in the genome and approximately 56% of the arginine codons in the transcriptome (*Gingold et al., 2012*). (**B**) Statistical significance of differences between proportions of discrepancies when masking the anticodon triplet versus control triplets. (a) Six control triplets were masked (*Figure 3—figure supplement 1*), and the analysis to identify switched tRNAs was repeated (see 'Materials and methods'). For each masked triplet, we computed the percentage of tRNA sequences that are not clustered according to their triplet content in each species. (b) The proportion of switched tRNAs was averaged across all species, and (c) the standard error of this average was computed. (d) The distribution of tRNA switching proportions from the original anticodon analysis was compared to the distribution of discrepancies from the control triplet, using a paired *t* test. The p value from this comparison is reported here. (**C**) Various yeast species tend to keep tRNAArgCCU in a single copy. All examined yeast species maintain a single copy of $tRNA^{Arg}_{CCU}$ compared to $tRNA^{Arg}_{UCU}$, which is mostly found in multiple copies. The copy number of $tRNA^{Arg}_{UCU}$ and $tRNA^{Arg}_{CCU}$ genes is shown together with the codon usage of AGG and AGA codons in the different yeast genomes.

• Supplementary file 2. Plasmids and primers.

### Major dataset

The following previously published datasets were used:

| Author(s) | Year | Dataset title | Dataset ID and/or URL | Database, license, and accessibility information |
|---|---|---|---|---|
| Chan PP, Lowe TM | 2009 | Genomic tRNA database | http://gtrnadb.ucsc.edu/ | We downloaded all the known tRNA sequences that are publicly available from the Genomic tRNA Database. |

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
