## [Decision Letter]

Thank you for sending your work entitled “tRNA genes rapidly change in evolution to meet novel translational demands” for consideration at *eLife*. Your article has been favorably evaluated by a Senior editor and 3 reviewers, one of whom is a member of our Board of Reviewing Editors.

The following individuals responsible for the peer review of your submission have agreed to reveal their identity: Michael Laub (Reviewing editor) and Chris Marx (peer reviewer).

The Reviewing editor and the other reviewers discussed their comments before we reached this decision, and the Reviewing editor has assembled the following comments to help you prepare a revised submission.

Overall, this paper was favorably received by all three reviewers. The general consensus was that the results provide important new insights into the evolution of tRNAs and represents a nice example of the power of experimental evolution to provide novel biological insights. There are, however, several concerns about the writing and interpretation of a few things that we felt warranted revision. More generally, there was a concern that the paper felt like a loose collection of vignettes and better writing to integrate them would serve the reader well. Each of these concerns about the writing must be dealt with through changes in the text. There is also one experiment that is straightforward that would be necessary to bolster one of the primary claims made in the text (item #1 below) and one experiment (item #6 below) that a reviewer suggests (it's not essential, but could really enhance the paper). All of the major issues to address, collated from the three independent reviewers, are listed below:

1) The authors argue that overproducing low-copy tRNAs causes proteotoxic stress. This conclusion rests primarily on a single assay involving the expression of an mCherry-tagged human protein. I think it's important to assess this by at least one other means; for example, testing whether various HSPs are induced (either individual HSPs or, preferably, a genome-wide transcriptional analysis by arrays or RNA-Seq).

(As a side note: it was not clear whether the potential proteotoxic stress observed resulted from misincorporation of amino acids per se or defects in co-translational folding that result from changes in translation speed/efficiency without misincorporation. The reviewers did not think experiments to address this were necessary for a revised manuscript, but we felt future experiments ought to try and distinguish between these models. For instance, one could measure rates of misincorporation. Or, if the defect is in cotranslational folding, one could imagine taking a protein in which a “slow” and a “fast” codon are near each other: it should be true (if cotranslational folding is the culprit) that making both codons “fast” is detrimental, but that swapping them (and retaining one slow, one fast) is tolerated. The misincorporation model would, by contrast, predict that the exact position of the “slow” codon matters.)

2) The authors often refer to “translation efficiency” in the text, but does this mean translation speed, accuracy, or some combination thereof? Can you be more explicit about the intended meaning?

3) The data are clear and compelling that “chimeric” tRNAs harboring the backbone of one tRNA and the anticodon of another can compete with wild-type under a range of conditions. But this begs the question of why duplicated tRNAs in yeast retain nearly 100% sequence identity - one would expect, given the data presented here, to see much more sequence diversity. Can this be commented on?

4) The first paragraph introduces codon bias, then states that the existence of codon usage implies that translation has undergone adaptation. This is not the case, because mutational biases are known to cause codon usage bias. The manuscript states that adaptation is “presumably” to fine-tune codon usage to gene expression. One need not presume, as the subject has been studied for decades. The authors should expand this section to give a more accurate picture of the question at hand. Plotkin & Kudla (Nat Rev Genet 2011) provide a solid review.

5) It has not been established that any tRNA scaffold with an anticodon change suffices to restore function and fitness, only isoaccepting scaffolds.

One would not expect the scaffold to be entirely interchangeable. Some aa-tRNA synthetases do not bind the anticodon (Leu, Ala and Ser-accepting aaRS), and the anticodon of the corresponding tRNAs can be changed without affecting charging (see Beuning and Musier-Forsyth, Biopolymers 1999). For these species, the scaffold is decisive for proper charging.

Thus, the claim that the tRNA scaffold does not confer tRNA functionality is incorrect. It also does not follow that the anticodon is the dominant feature. It may be the dominant feature so long as the tRNA scaffold is from a synonymous family.

6) Not reporting results after starving the cells for arginine seems a missed opportunity.

7) The levels of tRNAs achieved after multicopy plasmid expression remain unclear. At present, we do not know whether the fold-change in tRNA for each tRNA variant is equivalent, or whether the absolute change is equivalent. We also have no guidance as to which one matters. It would seem that if achieving an increase of x molecules of tRNA means a 10% increase in the abundant tRNA and a 10-fold increase in the rare tRNA, then the results obtained may not be surprising. Please address why adding a putatively constant amount of additional tRNA is the correct experiment.

8) The paper jumps to assuming codon usage bias emerges from selection before providing evidence that it isn't just random drift. Particularly for a broad audience like *eLife*, I think it is critical to distinguish that the existence of a bias is not enough to claim selection. It is the ways in which there are bias (such as between genes and within them, not just between organisms) that make it clear adaptation plays a role. This idea shows up at the end of the paragraph beginning “The fitness effects of an unmet translational demand…”, but by then adaptation has long since been assumed.

9) Related to the above point, the paragraph beginning “Furthermore, the forces that direct and maintain low copy tRNA families remain unclear…” is confusing as currently written. I think the point trying to be made is that, although on average selection favors a particular set of codons that are generally labeled as optimal, might there be an adaptive reason for low frequency/sub-optimal codons that – in certain circumstances – are actually selectively advantageous. Please try to clarify this point, as it is so central to the paper. Along these lines, the argument could be strengthened by citing one or more of the available papers that a gene encoded with 100% ‘optimal’ codons can actually be poorly expressed or cause decreased fitness.

10) An interesting point was made that detectable anticodon switching events occurred in genomes with a greater than average number of tRNA. Might there be ascertainment bias here? If such switches are easiest to detect for clades with many sequenced members, and these are generally fast pathogens with many tRNA (and rRNA copies, etc), then this may make at least some of this signal a simple consequence of where it was easiest to identify the events.

11) In their Discussion, they say, “our…search…revealed the prevalence of this adaptation mechanism in nature,” where I think they can at best say “It is not established that natural anticodon changes are adaptive, nor that, if they are adaptive, that they are a response to novel translational demands. Our search revealed the prevalence of tRNA anticodon mutations in nature, consistent with a response to novel translational demands as in our experimental evolution results”, or something of the sort.

---

## [Author Response]

*1) The authors argue that overproducing low-copy tRNAs causes proteotoxic stress. This conclusion rests primarily on a single assay involving the expression of an mCherry-tagged human protein. I think it's important to assess this by at least one other means; for example, testing whether various HSPs are induced (either individual HSPs or, preferably, a genome-wide transcriptional analysis by arrays or RNA-Seq)*.

Following the reviewers’ suggestion, we measured the expression level of six canonical heat shock proteins (HSPs) by RT-qPCR in order to establish the proteotoxic stress in the two tRNA overexpressing strains. Our new results (Figure 4) demonstrate that indeed HSP genes are induced 2–8-fold when overexpressing the originally low copy number tRNA gene. In contrast, none of the HSP genes was induced in the cells that overexpress the high copy number tRNA gene. Thus, the RT-PCR measurements support our initial claim that over-expressing a rare tRNA gene causes a proteotoxic stress, ascribing a pro-folding function to rare tRNAs, and suggest that at least part of the fitness reduction observed upon over expression of a rare tRNA is caused by proteotoxic stress.

*(As a side note: it was not clear whether the potential proteotoxic stress observed resulted from misincorporation of amino acids* per se *or defects in co-translational folding that result from changes in translation speed/efficiency without misincorporation. The reviewers did not think experiments to address this were necessary for a revised manuscript, but we felt future experiments ought to try and distinguish between these models. For instance, one could measure rates of misincorporation. Or, if the defect is in cotranslational folding, one could imagine taking a protein in which a “slow” and a “fast” codon are near each other: it should be true (if cotranslational folding is the culprit) that making both codons “fast” is detrimental, but that swapping them (and retaining one slow, one fast) is tolerated. The misincorporation model would, by contrast, predict that the exact position of the “slow” codon matters.*)

We agree that the proteotoxic stress may have several origins and thus discuss three mechanisms that may lead to it in the text. We also further clarify that the potential part of the observed proteotoxic stress due to mis-incorporation still remains to be studied.

Currently in our lab, we aim to pursue this line of research by protein sequencing. We use mass spectrometry technology on designated proteins enriched with arginine. We also thank the reviews for the suggested “slow” and “fast” adjacent arginine codons as it is an elegant experiment that can be performed in future by using the same mass-spec method.

*2) The authors often refer to “translation efficiency” in the text, but does this mean translation speed, accuracy, or some combination thereof? Can you be more explicit about the intended meaning*?

We now clarify in the text the definition of translation efficiency. We note that the translation efficiency affects both production throughput and accuracy.

*3) The data are clear and compelling that “chimeric” tRNAs harboring the backbone of one tRNA and the anticodon of another can compete with wild-type under a range of conditions. But this begs the question of why duplicated tRNAs in yeast retain nearly 100% sequence identity - one would expect, given the data presented here, to see much more sequence diversity. Can this be commented on*?

We agree that in light of our results the origin of the high sequence identity among tRNA genes is an interesting question. We added the following paragraph to the Discussion as follows:

“The present chimeric tRNAs that emerged in our lab-evolution experiments successfully replaced the deleted tRNA, despite differences in 20 nucleotide positions between the two tRNA scaffolds…”

*4) The first paragraph introduces codon bias, then states that the existence of codon usage implies that translation has undergone adaptation. This is not the case, because mutational biases are known to cause codon usage bias. The manuscript states that adaptation is “presumably” to fine-tune codon usage to gene expression. One need not presume, as the subject has been studied for decades. The authors should expand this section to give a more accurate picture of the question at hand. Plotkin & Kudla (Nat Rev Genet 2011) provide a solid review*.

We thank the reviewers for this important point. We extensively edited the first three paragraphs of the Introduction and elaborated on the evolutionary processes of codon usage bias.

*5) It has not been established that any tRNA scaffold with an anticodon change suffices to restore function and fitness, only isoaccepting scaffolds*.

One would not expect the scaffold to be entirely interchangeable. Some aa-tRNA synthetases do not bind the anticodon (Leu, Ala and Ser-accepting aaRS), and the anticodon of the corresponding tRNAs can be changed without affecting charging (see Beuning and Musier-Forsyth, Biopolymers 1999). For these species, the scaffold is decisive for proper charging.

*Thus, the claim that the tRNA scaffold does not confer tRNA functionality is incorrect. It also does not follow that the anticodon is the dominant feature. It may be the dominant feature so long as the tRNA scaffold is from a synonymous family*.

We agree that our observations focus on synonymous tRNA families only. Thus, we changed the text accordingly:

“Hence, the chimeric tRNA provides a direct in vivo indication that the scaffolds of tRNAs, which encode for the same amino-acid, may be interchangeable in terms of their effect on cellular growth under the conditions we tested.”

*6) Not reporting results after starving the cells for arginine seems a missed opportunity*.

We now preformed arginine-starving assays on the deletion strain (ΔtRNA^Arg^_CCU_) compared to the WT (not shown). Yet, we did not observe any significant difference in the growth, comparing this medium to a medium containing arginine. Accordingly, we did not pursue this line of research further.

*7) The levels of tRNAs achieved after multicopy plasmid expression remain unclear. At present, we do not know whether the fold-change in tRNA for each tRNA variant is equivalent, or whether the absolute change is equivalent. We also have no guidance as to which one matters. It would seem that if achieving an increase of x molecules of tRNA means a 10% increase in the abundant tRNA and a 10-fold increase in the rare tRNA, then the results obtained may not be surprising. Please address why adding a putatively constant amount of additional tRNA is the correct experiment*.

We acknowledge that this is a valid point, yet with the aim to recapitulates the natural evolutionary process of gene dosage changes we used similar increments in gene copy number for both rare and abundant tRNA. We started with addition of a centromeric plasmid (considered as an equivalent to addition of a single copy) and observed minor effects under severe growth conditions only. Therefore, we aimed to shift tRNA copies of both families to the extreme. For that we used high-copy plasmids in which both families were supplemented with the same number of copies. Since expression of each tRNA is determined by its endogenous promoter we could not control for differences in tRNA expression, only for differences in tRNA gene copy number. It is also worth noting that the copy number of the plasmid that we used (2μ plasmid) ranges from 20 to 80 in a single cell. Thus, we believe that the expression level of both rare and abundant tRNAs were greatly influenced in our experiments.

*8) The paper jumps to assuming codon usage bias emerges from selection before providing evidence that it isn't just random drift. Particularly for a broad audience like* eLife*, I think it is critical to distinguish that the existence of a bias is not enough to claim selection. It is the ways in which there are bias (such as between genes and within them, not just between organisms) that make it clear adaptation plays a role. This idea shows up at the end of the paragraph beginning “The fitness effects of an unmet translational demand…”, but by then adaptation has long since been assumed*.

We thank the reviewers for this important point and we have now edited the first three paragraphs of the Introduction to clarify the role of selection and drift in the phenomenon of codon usage bias.

*9) Related to the above point, the paragraph beginning “Furthermore, the forces that direct and maintain low copy tRNA families remain unclear…” is confusing as currently written. I think the point trying to be made is that, although on average selection favors a particular set of codons that are generally labeled as optimal, might there be an adaptive reason for low frequency/sub-optimal codons that* – *in certain circumstances* – *are actually selectively advantageous. Please try to clarify this point, as it is so central to the paper. Along these lines, the argument could be strengthened by citing one or more of the available papers that a gene encoded with 100% ‘optimal’ codons can actually be poorly expressed or cause decreased fitness*.

We have revised this paragraph completely in order to express the point in a clearer fashion. We also added published examples, according to the reviewers’ suggestion, that demonstrate how “non-optimal codons” have advantageous functions in translation. The new paragraph now reads as follows:

“Further, the forces that direct and maintain low copy tRNA families remain unclear…”

*10) An interesting point was made that detectable anticodon switching events occurred in genomes with a greater than average number of tRNA. Might there be ascertainment bias here? If such switches are easiest to detect for clades with many sequenced members, and these are generally fast pathogens with many tRNA (and rRNA copies, etc), then this may make at least some of this signal a simple consequence of where it was easiest to identify the events*.

Indeed, we observed that genomes with large tRNA pools demonstrate more tRNA switching events. However, thanks to the reviewer comments, we noticed that there are no simple relations between the number of tRNA genes and the number of tRNA switching events per species. We thus took out our original statement about the difference between the species and merely state that this issue warrants future analysis.

Additionally, it is worth noting that our tRNA switching analysis was performed on individual species and not clades and that all tRNAs are discovered by scanning whole genomic sequences with the tRNA-SCAN software and then scored with the COVE algorithm for the likelihood of being a tRNA gene. Thus, even though there might be a bias in the available genomic data (the species sequenced so far) our screen is able to detect most of the switching event per species.

We thank the reviewers for raising this important point and clarify the issue in the text of both the Results and Materials and methods sections.

*11) In their Discussion, they say, “our…search…revealed the prevalence of this adaptation mechanism in nature,” where I think they can at best say “It is not established that natural anticodon changes are adaptive, nor that, if they are adaptive, that they are a response to novel translational demands. Our search revealed the prevalence of tRNA anticodon mutations in nature, consistent with a response to novel translational demands as in our experimental evolution results”, or something of the sort*.

We changed the text according to the reviewers’ comment by modifying the end of the paragraph on as follows:

“Furthermore, our systematic search for tRNA switching events throughout the tree of life revealed the prevalence of tRNA anticodon mutations in nature. This observation is consistent with the results of our lab-evolution experiment and may be the evolutionary outcome to novel translational demands in the wild.”